# Tryptophan Metabolism, Inflammation, and Oxidative Stress in Patients with Neurovascular Disease

**DOI:** 10.3390/metabo10050208

**Published:** 2020-05-19

**Authors:** Martin Hajsl, Alzbeta Hlavackova, Karolina Broulikova, Martin Sramek, Martin Maly, Jan E. Dyr, Jiri Suttnar

**Affiliations:** 1Department of Medicine, First Faculty of Medicine, Charles University in Prague and the Military University Hospital, 16902 Prague, Czech Republic; martin.hajsl@uvn.cz (M.H.); karolina.broulikova@uvn.cz (K.B.); martin.maly@uvn.cz (M.M.); 2Department of Military Internal Medicine and Military Hygiene, Faculty of Military Health Sciences, University of Defence, 50002 Hradec Kralove, Czech Republic; 3Department of Biochemistry, Institute of Hematology and Blood Transfusion, 12820 Prague, Czech Republic; alzbeta.hlavackova@uhkt.cz (A.H.); jan.dyr@uhkt.cz (J.E.D.); 4Department of Neurosurgery and Neurooncology, First Faculty of Medicine, Charles University in Prague and the Military University Hospital, 16902 Prague, Czech Republic; martin.sramek@uvn.cz

**Keywords:** tryptophan metabolism, atherosclerosis, acute ischemic stroke, carotid artery stenosis, inflammation, oxidative stress

## Abstract

Atherosclerosis is a leading cause of major vascular events, myocardial infarction, and ischemic stroke. Tryptophan (TRP) catabolism was recognized as an important player in inflammation and immune response having together with oxidative stress (OS) significant effects on each phase of atherosclerosis. The aim of the study is to analyze the relationship of plasma levels of TRP metabolites, inflammation, and OS in patients with neurovascular diseases (acute ischemic stroke (AIS), significant carotid artery stenosis (SCAS)) and in healthy controls. Blood samples were collected from 43 patients (25 with SCAS, 18 with AIS) and from 25 healthy controls. The concentrations of twelve TRP metabolites, riboflavin, neopterin (NEO, marker of inflammation), and malondialdehyde (MDA, marker of OS) were measured by liquid chromatography–tandem mass spectrometry (LC-MS/MS). Concentrations of seven TRP metabolites (TRP, kynurenine (KYN), 3-hydroxykynurenine (3-HK), 3-hydroxyanthranilic acid (3-HAA), anthranilic acid (AA), melatonin (MEL), tryptamine (TA)), NEO, and MDA were significantly different in the studied groups. Significantly lower concentrations of TRP, KYN, 3-HAA, MEL, TA, and higher MDA concentrations were found in AIS compared to SCAS patients. MDA concentration was higher in both AIS and SCAS group (*p* < 0.001, *p* = 0.004, respectively) compared to controls, NEO concentration was enhanced (*p* < 0.003) in AIS. MDA did not directly correlate with TRP metabolites in the study groups, except for 1) a negative correlation with kynurenine acid and 2) the activity of kynurenine aminotransferase in AIS patients (*r* = −0.552, *p* = 0.018; *r* = −0.504, *p* = 0.033, respectively). In summary, TRP metabolism is clearly more deregulated in AIS compared to SCAS patients; the effect of TRP metabolites on OS should be further elucidated.

## 1. Introduction

Atherosclerotic vascular diseases (ASVD) are the leading cause of death worldwide. The most severe forms of ASVD are acute coronary syndrome and acute ischemic stroke, which are caused by superimposed thrombosis on underlying atherosclerotic plaque. The pathogenesis of atherosclerosis closely connects activation of proinflammatory signaling pathways, expression of cytokines/chemokines, and production of reactive oxygen species (ROS) [1,2,3]. Both immune and inflammatory responses together with oxidative stress (OS) have significant effects on every phase of atherosclerosis [4,5]. 

In recent years, tryptophan (TRP) catabolism via the kynurenine (KYN) pathway (KP) was recognized as an important player in inflammation and immune response [6,7,8,9,10]. The KP accounts for about ~95% of overall TRP degradation to bioactive catabolites, collectively called kynurenines forming the primary end product NAD+. Three key enzymes of KP, tryptophan 2,3-dioxygenase (TDO), and indoleamine 2,3-dioxygenase 1 and 2 (IDO-1, IDO-2) have been described [11,12]. TDO was found to catalyze TRP breakdown under basal conditions, while IDO-1, with key role in immune regulation, is induced and regulated by several stimuli, such as inflammatory signals [13]. 

IDO-1 expression is induced by cytokine interferon γ (INF-γ) [9]. INF-γ activity is reflected by plasma neopterin (NEO) and KYN: TRP (KTR) ratio that are correlated and often considered systemic inflammatory markers and markers of IDO-1 activity [14,15,16]. Enzyme activities are usually expressed in the TRP literature as ratios (or ratio percentages) of products to substrates [17]. The major TRP metabolic pathways in humans are shown in Figure 1.

KP is sensitive to changes in the concentration of B vitamins that play a crucial role as cofactors. The active form of vitamin B6 (pyridoxal 5’-phosphate, PLP) affects kynureninase (KYNU) and kynurenine aminotransferase (KAT). Lowered vitamin B2 concentration results in a reduction of the activity of the flavin adenine dinucleotide dependent kynurenine 3-monooxygenase (KMO) [18,19]. B-vitamins, including riboflavin (RBF) [20] and pyridoxin [21], play a role in the prevention of stroke and in the recovery after stroke. 

It was reported that abnormal KP is involved in neurological disorders [22], cancer [8], cardiovascular diseases (CVD) [23], and in stroke [24]. One of the risk factors for stroke is carotid artery stenosis [25,26] with considerably fewer references to tryptophan metabolism [27]. Stenosis of the carotid artery and ischaemic stroke is herein represented as neurovascular disease.

KYN metabolites in the main branch of KP (3-hydroxykynurenine (3-HK), 3-hydroxyanthranilic acid (3-HAA), quinolinic acid (QA), and picolinic acid (PA)) are neuroactive compounds with immunomodulatory effects [12]. Several kynurenines, including KYN, kynurenic acid (KA), anthranilic acid (AA), 3-HK, xanthurenic acid, 3-HAA, and cinnabarinic acid produced from 3-HAA by catalase as a minor product are endogenous ligands for the aryl hydrocarbon receptor (AhR) taking part in the regulation of innate and adaptive immune cell functions [28,29]. Kynurenine/AhR pathway mediates acute brain damage after stroke [30]. Kynurenine metabolites also affect N-methyl-D-aspartate (NMDAR) receptor mediating stroke injury, where its antagonist KA is generally considered neuroprotective and its agonist QA is excitotoxic [31,32].

In addition, high concentrations of kynurenine metabolites may be either prooxidative or antioxidant and thereby have the potential to generate or scavenge ROS [33]. As an example, in a study where 15 human subjects were overloaded by oral administration of 6 g TRP, induction of OS was observed probably because of the generation of KP metabolites QA, 3-HA, and 3-HAA, all of which are known to have the ability to generate free radicals [34]. Accordingly, the inhibition of the KP prevented OS induced in the brain of adult rats by ketamine [35]. Reliable markers of OS are carbonylated proteins, malondialdehyde (MDA), 4-hydroxy-2-nonenal, and F2-isoprostanes analyzed mostly by LC-MS/MS [36].

About 5% of TRP is metabolized to serotonin (5-HT), 5-hydroxyindoleacetic acid (5-HIAA), melatonin (MEL), and to tryptamine (TA). Recently, 5-HT was identified as an activator of immune responses and inflammation [37,38] besides its role in neurotransmission, vasoconstriction, or vasodilation of blood vessels, control of hemostasis and platelet function [39]. MEL is frequently mentioned in connection with bodily rhythms and as an antioxidant lowering the OS [40]. Recently, it was shown that MEL is also involved in the interactions between the nervous, endocrine, and immune systems and is a beneficial agent in the treatment of inflammatory and immune disorders [41,42] including CVD and neurovascular diseases. 

Since TRP catabolism was recognized as an important player in inflammation and its metabolites with either prooxidant and antioxidant properties should have relationship with OS, in this study we aim to compare the similarities and differences in TRP metabolism, inflammation, and OS in patients with acute ischemic stroke (AIS), more rarely studied significant carotid artery stenosis (SCAS) and in healthy controls. The blood plasma concentrations of both KP and serotonin pathway metabolites (TRP, KYN, 3-HK, 3-HAA, KA, AA, QA, PA, 5-HT, 5-HIAA, MEL), tryptamine (TA), markers of inflammation (NEO, KTR), OS marker MDA, and RBF as one cofactor of enzymes in KP were analyzed and their mutual correlations in studied groups of probands were explored.

## 2. Results

Concentrations of TRP metabolites, RBF, NEO, MDA, values of KTR, and KP enzyme activities expressed as ratio percentages of products to substrates are shown in Table 1.

### 2.1. Metabolites of KP

Five metabolites of KP with significantly different concentrations were found in patients with SCAS, AIS and in the control group: TRP, KYN, 3-HK, 3-HAA, and AA. The 3-HAA/AA ratio and KTR were also different (*p* < 0.001, *p* = 0.003, respectively). Box plots revealed significant differences in the concentrations of TRP, KYN, and 3-HAA in patients with AIS compared to patients with SCAS. The patients with SCAS significantly differed in concentrations of TRP, 3-HK, AA, and in HAA/AA when compared with the controls. The patients with AIS and the controls significantly differed in concentrations of TRP, 3-HK, 3-HAA, AA, and in HAA/AA (Figure 2).

The IDO activity expressed as KTR was significantly different in SCAS patients compared to controls (*p* = 0.002). The other enzyme activities expressed as ratio percentages of products to substrates were significantly different (*p* < 0.001) for 100 × [3-HK]/[KYN] (KMO activity), 100 × [AA]/[KYN] (KYNU A activity), 100 × [3-HAA]/[HK] (KYNU B activity), 100 × [QA]/[3-HAA] (3-HAO activity), and for 100 × [PA]/[3-HAA] (mixed activity of 3-HAO and 2-amino-3-carboxymuconate-semialdehyde decarboxylase (ACMSD)) in the studied groups of patients (Figure 3). 

### 2.2. Non Kynurenines Metabolites

5-HT, MEL, 5-HIAA, and TA as non-KP TRP pathways metabolites were analyzed in plasma patients with neurovascular disease and controls. Both MEL and TA concentrations were significantly different in controls and in patients with SCAS or AIS (*p* < 0.001, *p* = 0.001, respectively). (Figure 4).

Patients with AIS and patients with SCAS significantly differed in the concentrations of MEL and TA (*p* = 0.045, *p* = 0.02, respectively). MEL and TA strongly positively correlated (*r* = 0.7–0.9, *p* < 0.001) in all groups of patients. Both 5-HIAA and 5-HT plasma concentrations were not significantly different in patients groups of study.

### 2.3. Markers of Inflammation, OS, and RBF

MDA was employed as a criterion of OS, inflammation was assessed by measuring the plasma NEO concentration and KTR. RBF was measured as a source of FAD, coenzyme of KMO. Both MDA and NEO concentrations, and KTR values were significantly different in the controls and in the patients with SCAS or AIS (*p* < 0.001, *p* = 0.004, *p* = 0.003, respectively) (Table 1). Significantly higher MDA concentration was found in the plasma of patients with AIS when compared with the plasma of patients with SCAS (*p* = 0.008) (Figure 5a). The correlations of MDA with TRP metabolites were found only in AIS patients (Figure A1) where MDA moderately negatively correlated with both KA and activity of KAT (100 × [KA]/[KYN]) (*r* = −0.552, *p* = 0.018; *r* = −0.504, *p* = 0.033, respectively). 

The concentration of NEO in plasma of patients with neurovascular disease was enhanced compared with controls, but the significant enhancement was observed only in the plasma of patients with AIS (*p* = 0.003) (Figure 5b). Similarly, KTR was enhanced in plasma of patients with neurovascular disease compared with controls, but the significant enhancement was observed only in the plasma of patients with SCAS (*p* = 0.002). 3-HK moderately positively correlated with inflammatory marker NEO (*r* = 0.403, *p* = 0.046; *r* = 0.631, *p* = 0.005, respectively) and KTR (*r* = 0.526, *p* = 0.007; *r* = 0.490, *p* = 0.04, respectively) in patients with SCAS and AIS (Figure A2a,b). QA moderately positively correlated with 3-HK (*r* = 0.523, *p* = 0.007; *r* = 0.640, *p* = 0.004, respectively) and KTR (*r* = 0.684, *p* < 0.001; *r* = 0.621, *p* = 0.006, respectively) in patients with SCAS and AIS (Figure A2c,d). QA also moderately positively correlated with NEO (*r* = 0.595, *p* = 0.002) in patients with SCAS (Figure A3a). Strong positive correlation of KTR with NEO concentrations was found in the plasma of patients with SCAS and a moderate positive correlation in plasma of all probands (*r* = 0.81, *p* < 0.001, *r* = 0.657, *p* < 0.001, respectively) (Figure A3b).

RBF plasma concentrations were not significantly different in patients groups of study, nevertheless RBF moderately positively correlated with KMO activity (*r* = 0.554, *p* = 0.017) in patients with AIS.

## 3. Discussion

In this work patients with SCAS, AIS, and controls were compared with respect to TRP metabolism, inflammation, OS, and RBF.

Overall, Kruskal–Wallis non-parametric tests revealed significant differences of five TRP metabolites concentrations in KP together with changes of involved enzymatic activities in studied groups of patients. Moreover, concentrations of two of four measured metabolites in serotonin pathway were significantly changed. The inflammation as well as OS was enhanced both in SCAS and AIS patients compared to control.

TRP levels in neurovascular patients were significantly lower as compared with controls, with significantly lower levels of TRP in AIS patients compared to patients with SCAS. The result is in accordance with systematic Colpo’s review evaluating the involvement of KP in stroke [24] and with Wang’s review about dysregulation of KP in inflammation [27]. Accordigly, neurovascular patients had both enhanced inflammation marker NEO (AIS patients significantly) and KTR ((SCAS patients significantly) as compared with controls.

Since KTR is considered also as the IDO activity criterion [16], higher concetration of KYN and other KP metabolites in patients with neurovascular disease was expected compared with controls. However, KYN concentrations in neurovascular patients were not significantly different compared to controls, nevertheless KYN concentrations in SCAS patients were significantly higher compared to AIS patients. The results do not correspond neither to Colpo’s review for stroke [24] nor to Pawlak’s results about KYN and QA linked to carotid atherosclerosis in patients with end-stage renal disease [43]. However, Ormstad et al. also did not found significant differences of both KYN levels and KTR values in patients with AIS [44]. Similarly, in Mo study KYN levels were not significantly changed while KTR levels were significantly enhanced in stroke patients [45].

Recently, it was shown that peripheral KTR is influenced by various factors and is not a reliable marker of IDO activity [16,46]. KTR is reliable for measurement of IDO activity in isolated cells or cell cultures of cells capable of producing functional IDO, e.g., in endothelial cells, antigen-presenting cells, fibroblasts, macrophages, and dendritic cells [10,46]. In plasma, IDO-independent loss of TRP (enhancing KTR) may be caused by decreased dietary intake of TRP or due to the action of liver TDO increase for instance in renal diseases [43]. Finally, the upregulation of KMO and KYNU activities can lead to the decrease of circulating and tissue KYN and thus lower KTR [16]. Indeed, the levels of 3-HK, AA, and KYNU A activity in neurovascular patients were significantly enhanced compared to controls. Moreover, the KMO activity in AIS patients was significantly higher compared both to controls and SCAS patients. This can explain both the lower KYN concentration and lower KTR in AIS patients relative to SCAS patients.

Enhanced NEO in AIS patients reflects enhanced levels of INF-γ, one of cytokines that besides IDO also upregulates KMO, KYNU, and 3-hydroxyanthranilate 3,4-dioxygenase (3-HAO) activities in KP [9]. Furthermore, KMO activity depends on vitamin B2 concentration [18,19]. RBF concentration in neurovascular patients was unchanged, but it was positively correlated with KMO activity in patients with AIS. Therefore, enhanced KMO activity in AIS patients can be attributed to higher levels of NEO (INF-γ). NEO levels were also shown as a predictor strongly associated with major adverse clinical outcome in patients after AIS [47].

3-HK in neurovascular patients was positively correlated with inflammation markers (NEO, KTR), and QA, similarly as was reported for patients with CVD and cancer [15]. It was shown, that both 3-HK and QA are immunomodulators and are connected with inflammation [12]. It is a controversial metabolite that acts either as a free radical scavenger or as a producer of ROS depending on the reaction conditions [33]. The enhanced levels of 3-HK in neurovascular patients compared to controls were accompanied with enhanced OS measured as total MDA levels in plasma. Nevertheless, 3-HK did not significantly correlate with MDA in either group of patients.

AA level as well as KYNU A activity was enhanced in neurovascular patients compared to controls. Enhanced AA levels could be elucidated besides increased KYNU A activity by inhibition of an either hypothetical AA oxidase in brain or cytochrome P-450 in liver [48,49]. AA is generally considered biologically inactive, however it has antioxidant properties implemented either by complexing with iron ions or by direct removal of ROS [33].

Another important TRP metabolite in the main KP is 3-HAA formed from 3-HK by KYNU B activity. 3-HAA has multifaceted functions. It suppresses cytokine and chemokine production, and has antioxidant properties [50,51,52]. In the presence of metal ions it exhibits pro-oxidant behavior [53].

Neurovascular patients were characterized by significantly lowered concentration of 3-HAA in AIS patients compared with SCAS patients and controls. Both SCAS and AIS patients had lowered ratio of 3-HAA to AA as well as the lowered levels of KYNU B activity as compared with controls. Our result is in contrast with the enhanced values of 3-HAA observed in myocardial infarction [23] but in complete agreement with outcome of Darlington’s experiments dealing with altered kynurenine metabolism in stroke [54]. Darlington revealed that the significant decrease in the ratio of 3-HAA/AA was strongly correlated with infarct volume. He hypothesized that the levels of 3-HAA or the values of 3-HAA/AA may contribute to disorders with an inflammatory component, and may represent a novel marker for the assessment of inflammation and its progression. Decreased levels of 3-HAA with simultaneously elevated AA levels have been commonly observed in a variety of neurological and other disorders, including chronic brain damage, Huntington’s disease, ischemic heart disease, stroke, and depression [55]. Darlington explained the changes in 3-HAA and AA levels mainly by inhibition of hypothetical AA oxidase in the brain. However, AA oxidase activity in brain leading to production of 3-HAA can scarcely influence the concentration of 3-HAA in plasma, since 3-HAA crosses the blood-brain barrier only poorly [56]. Main part of 3-HAA is metabolized to QA and PA. QA concentrations tended to higher values in neurovascular patients as compared to control whereas PA concentrations were not changed. Nevertheless, the activity of 3-HAO in neurovascular patients as well as combined activity 3-HAO and ACMSD in AIS patients was significantly enhanced as compared with controls. Thus the lowered concentrations of 3-HAA in neurovascular patients could be mainly attributed to higher rates of its transformation to QA and PA.

Analysis of non-kynurenine TRP metabolites revealed a two-fold nonsignificant increase in plasma levels of 5-HT in SCAS patients compared to patients with AIS and controls. This could correspond to 5-HT release from platelets because of partial platelets activation on atherosclerotic plaques in SCAS patients. Enhanced concentrations of MEL were found in the plasma of the patients with neurovascular disease when compared with the controls. The MEL plasmatic concentrations were even significantly higher in SCAS patients as compared with AIS patients. The enhanced MEL concentrations were surprising since we supposed their lowering due to enhanced OS. The lowered MEL concentrations were observed previously in patients with CVD [57]. On the other hand, Tan et al. proposed a hypothesis about provoking of melatonin synthesis by ROS, thus serving as a protection against the potential oxidative injury induced by the physiological ischemia/reperfusion [58].

The patients with neurovascular disease were not treated with MEL, therefore the increased levels of MEL could be paradoxically in a connection with the enhanced OS. Nevertheless, plasma concentrations of MEL depend on dynamic circadian rhythms [40]. Since the median time of the blood-drawing of either group of patients was significantly different, we were not able to assign the enhanced MEL concentrations unambiguously to the neurovascular disease.

The concentrations of a trace amine neurotransmitter TA (the neuromodulator of 5-HT) were significantly enhanced in SCAS patients as compared with controls and AIS patients. Since TRP readily crosses blood-brain barrier by large neutral amino acids transporter, plasmatic TRP concentration is an important factor controlling the synthesis of 5-HT, MEL, and TA in human brain in normal conditions [59]. MEL and TA easily cross the blood-brain barrier, therefore, higher plasma concentrations of TA and MEL in SCAS patients compared to AIS patients may be due to the combined effect of higher TRP concentrations and lower OS in SCAS patients.

Neurovascular patients had significantly higher OS as compared with controls. The levels of plasmatic MDA were even significantly higher in AIS patients compared to SCAS patients. The high MDA levels in AIS patients compared to controls were accompanied by lowered concentrations of 3-HAA. Lowered concentration of 3-HAA found in the patients with AIS when compared both with the SCAS patients and controls may indicate its antioxidant role [51,52]. Since more TRP metabolites have either antioxidant or prooxidant properties we supposed their correlations with MDA. Nevertheless, only MDA in AIS patients significantly negatively correlated with both KA concentration and activity of KAT agreeing with the role of KA in ROS scavenging [33].

Our study had some limitations. The number of probands in studied groups was relatively small. There is a possibility of connection between dietary tryptophan consumption and risk for carotid artery stenosis and stroke. However, we had no information about dietary habits of the patients. We measured only one marker of oxidative stress: MDA. Nevertheless, MDA measured by LC-MS/MS is a reliable commonly used marker of OS status. Measurement of another inflammatory marker, such as a C-reactive protein, could be useful to confirm the increased inflammation in neurovascular patients. Since some enzymes in KP are vitamin B6-dependent (KYNU, KAT), PLP measurements could refine information about changes in KP. IDO-1 is strongly inhibited by nitric oxide (NO) produced by inducible NO-synthase (iNOS) [6]. It is also known that concentrations of an iNOS inhibitor asymmetric dimethylarginine are enhanced in CVD and stroke. It would be interesting to measure the concentrations of methylated arginines in connection with TRP metabolism.

In conclusion, we revealed significant differences between SCAS and AIS patients in TRP metabolism, inflammation, and OS. For the first time, it was found that the 3-HAA/AA ratio in SCAS patients was reduced compared to controls and was comparable with values for AIS patients. OS in AIS patients was significantly higher compared to SCAS patients and controls. Nevertheless, no correlation was found between the marker of OS MDA and metabolites of TRP in the study groups, except for a negative correlation with antioxidant KA and the activity of KAT in AIS patients. Moderate correlations of TRP metabolites with NEO and KTR in patients with SCAS and AIS clearly supported the relationship of inflammation and TRP metabolism. The results showed that TRP metabolism was more deregulated in AIS compared to SCAS patients and that the effect of TRP metabolites on OS should be further clarified.

## 4. Materials and Methods

### 4.1. Sample Collection and Preparation

We have decided to use the clinical manifestation as a selection criterion for three groups. Patients with SCAS represents from pathophysiological point of view the “atherosclerotic” cohort. The stenosis is due to atherosclerotic plaque, and therefore the presence of advanced atherosclerosis is well defined. While the stenosis of the carotid artery represents the atherosclerosis, the AIS patients represents the “thrombotic” cohort. Controls were healthy elderly blood donors.

Blood samples were collected from 43 patients (25 with SCAS, 18 with AIS), diagnosed in the Military University Hospital, Prague, Czech Republic (Table 2). The patients with carotid artery stenosis were diagnosed using ultrasound carotid artery stenosis screening. The severity of the carotid artery stenosis was 81 ± 12% (mean ± SD). The patients with AIS were consecutive patients indicated for endovascular treatment with large vessel occlusion up to 6 h from the onset of the stroke.

Control samples were collected from 25 healthy elderly blood donors in the Institute of Hematology and Blood Transfusion, Prague, Czech Republic. Because of the different subgroup, only the basic available characteristics of the control group are listed here. The median age of the controls was 59 (38.2–65.0) (5th–95th percentiles in parentheses), female/male 11/14. All of the tested individuals agreed to participate in the study based on informed consent. All samples were obtained and analyzed in accordance with the Ethical Committee regulations of the Military University Hospital, Prague, Czech Republic (108/11–49/2017) and the Ethical Committee regulations of the Institute of Hematology and Blood Transfusion, Prague, Czech Republic (EK 9/AZV CR/06/2017). The study was carried out in accordance with the International Ethical Guidelines and the Declaration of Helsinki.

Blood samples were drawn from the patients and controls in vacutainer tubes containing EDTA, and centrifuged immediately at 4000× *g* for 5 min at 4 °C. The obtained plasma samples were stored in the dark at −80 °C until the analysis. The time of the blood-drawing of the groups of patients was different, with a median of 7:30 a.m. for the patients with SCAS, 0:36 p.m. for the patients with AIS, and 9:48 a.m. for the controls.

### 4.2. Reagents

All of the chemicals were obtained from Sigma-Aldrich (Prague, Czech Republic) unless otherwise specified. Chromatographic solvents were from Merck (Prague, Czech Republic). All of the reagents employed were of analytical grade or higher purity, and all aqueous solutions were prepared using HPLC-grade water.

### 4.3. Total MDA Analysis

Total malondialdehyde concentration in plasma was determined after alkaline hydrolysis with sodium hydroxide using liquid chromatography–tandem mass spectrometry (LC-MS/MS) method based on the Sim [60] and Mendonça procedure [61].

Both MDA and internal standard (IS) 2-methylmalondialdehyde (MetMDA) stock solutions were prepared by acid hydrolysis of 1,1,3,3-tetraethoxypropane or 1,1,3,3-tetraethoxy-2-methylpropane (Seratec, Göttingen, Germany) according to Suttnar [62].

MDA calibration standards were prepared in the range 30–0 μM (which means 3–0 μM in plasma) by sequential dilution of solutions. Total of 100 μL of plasma samples were mixed with 10 μL of IS (20 μM, IS), 10 μL calibration standards for seven point calibration, and 10 μL H_2_O for patient plasma samples, respectively, 5 μL 2,6-ditert-butyl-4-methylphenol (BHT) (150 mM) and 25 μL NaOH (6 M). Samples were incubated for 30 min at 60 °C. After incubation, 100 μL HClO_4_ (3 M) were added and samples were centrifuged 37,000× *g* for 10 min at 5° C. Total of 150 μL of supernatant was derivatized by adding 15 μL DNPH (5 mM) for 30 min in the dark. Finally, samples were centrifuged at 37,000× *g* for 30 min at 5 °C. 20 µl of the supernatant was injected into the HPLC column Nucleosil C18 ec (125 × 3 mm, 5 μm) (Macherey-Nagel, Düren, Germany) at 40 °C using an isocratic mobile phase composed of 0.1% of formic acid in 50% acetonitrile (v/v). The flow rate was 0.4 mL/min.

The HPLC system was connected to the mass spectrometer QTRAP 4000 (Sciex, Prague, Czech Republic). MDA and MetMDA DNPH derivatives (MDA-DNPH and MetMDA-DNPH) were detected in positive multiple reaction monitoring (MRM) mode. The ion source was operated using ion spray voltage set at 5500 V, curtain gas at 25 psi, ion source temperature at 500 °C, ion source gas 1 at 40 psi, gas 2 at 60 psi, and collision gas at medium. MDA-DNPH was monitored at *m/z* 235→189, declustering potential voltage (DP), 66 V, collision energy (CE), 23 V; collision cell exit potential (CXP), 20V and entrance potential (EP), 10 V. MetMDA-DNPH was monitored at *m/z* 249→203, DP, 91 V; CE, 25 V; CXP, 14 V and EP, 10 V. DNPH derivatives of MDA and MetMDA eluted at 3.05 and at 3.54 min, respectively; total time of analysis was 6.5 min. All MS parameters were optimized by direct infusion; the source parameters, by flow injection. Analyst v.1.6 from SCIEX was used for the acquisition and analysis of data.

### 4.4. Tryptophan Metabolites, NEO and RBF Analysis

The tryptophan metabolites and neopterin in plasma were analyzed essentially according to Zhu et al. [63]. Six different concentrations of calibration standards were prepared by sequential dilution of solutions. The concentration ranges of individual analytes in calibration mixture were for 5-HT 20–0 [μM]; AA 1–0 [μM]; 3-HAA 2–0 [μM]; 5-HIAA 4–0 [μM]; 3-HK 2–0 [μM]; KA 2–0 [μM]; KYN 200–0 [μM]; MEL 0.2–0 [μM]; NEO 1–0 [μM]; picolinic acid (PA) 2–0 [μM]; QA 10–0 [μM]; TRP 2000–0 [μM]; TA 0.3–0 [μM], RBF 0.4–0 [μM].

Total of 50 μL of plasma sample was mixed with 10 μL of 6-fluoro-L-tryptophan (6-F-TRP) used as an internal standard (40 μM, 6-F-TRP (IS)) and 50 μL of 0.1% formic acid (*v/v*). Five μL of calibration standards was added for calibration (Cal 1–6) and 5 μL H_2_O for patient plasma samples, respectively. Samples were deproteinized and extracted by adding of 440 μL ice-cold methanol. After vortex mixing all samples were incubated for 1 h at –25 °C and centrifuged at 37,000× *g* for 10 min at 5 °C. 450 μL of supernatants were placed in the new tubes, dried in a centrifugal vacuum concentrator Savant SPD 131 (Fischer Scientific, Pardubice, Czech Republic) and redissolved in 50 μL 0.1% formic acid (*v/v*). Finally, samples were centrifuged at 37,000× *g* for 30 min at 5 °C. The prepared samples including calibrations were ten times diluted for TRP measurement.

HPLC analysis was performed using a Prominence HPLC system (Shimadzu, Prague, Czech Republic). Total of 20 µl of sample was injected onto an Atlantis T3 column (150 × 2.1 mm, 5 μm) (Waters, Milford, MA, USA). The column was kept at 40 °C, the flow rate was 0.25 mL/min. The mobile phases were composed of 0.1% formic acid (*v/v*) (A) and 0.1% formic acid/acetonitrile (*v/v*) (B). Chromatographic separation of the analytes was performed using a linear gradient as follows: *t* (min)/% B: 0/0, 2/15, 6/26, 7.5/90, 11.8/90, 12/0. Total run time was 20 min. The HPLC system was connected to the mass spectrometer QTRAP 4000. The ion source was operated with ion spray voltage set at 5500 V, curtain gas at 25 psi, ion source temperature at 500 °C, ion source gas 1 and 2 at 45 psi, and collision gas at medium. The analytes were detected using positive MRM mode applying the parameters listed in Table A1. All MS parameters were optimized by direct infusion; the source parameters, by flow injection. Analyst v.1.6 from SCIEX was used for the acquisition and analysis of data.

Validation experiments of the method were performed. The linear range for each analyte was evaluated using a series of diluted calibration mixtures with internal standard added to each sample measured five times. The calibration curves were constructed using ratios of the analytes to internal standard versus the corresponding concentration of analyte. The solutions were prepared in 4% bovine albumin in phosphate buffered saline (Sigma, Prague, Czech Republic) treated with 15 mg/mL of active charcoal (Sigma, Prague, Czech Republic) overnight at room temperature. Signal to noise (S/N) value for each analyte was evaluated using the script within Analyst software. The LOD for each analyte was defined as concentration of analyte with S/N ≥ 3. The results are summarized in Table A2.

Matrix effects were calculated as a percentage of ratio of calibration curve slope in plasma and calibration curve slope in aqueous solution [64]. Six different plasma samples were spiked with analytes in a range of concentrations indicated in Table A2.

The recovery and imprecision was determined in six different plasma samples spiked with calibration mixture at concentrations indicated in Table A3. The recovery was calculated as a percentage of difference of measured and added concentration divided by added concentration.

### 4.5. Statistical Analysis

The statistical analysis was performed using R software [65]. The Shapiro–Wilk test of normality was used for data distribution analysis. The measured variables were not normally distributed, therefore a nonparametric Kruskal–Wallis test was used to examine the differences across all subgroups of patients. When the Kruskal–Wallis test was statistically significant, a post-hoc Dunn test was performed. Correlations between measured quantities were performed using a Pearson correlation test. All tests for statistical significance were standardized at an alpha level of *p* < 0.05.

## Figures and Tables

**Figure 1 metabolites-10-00208-f001:**
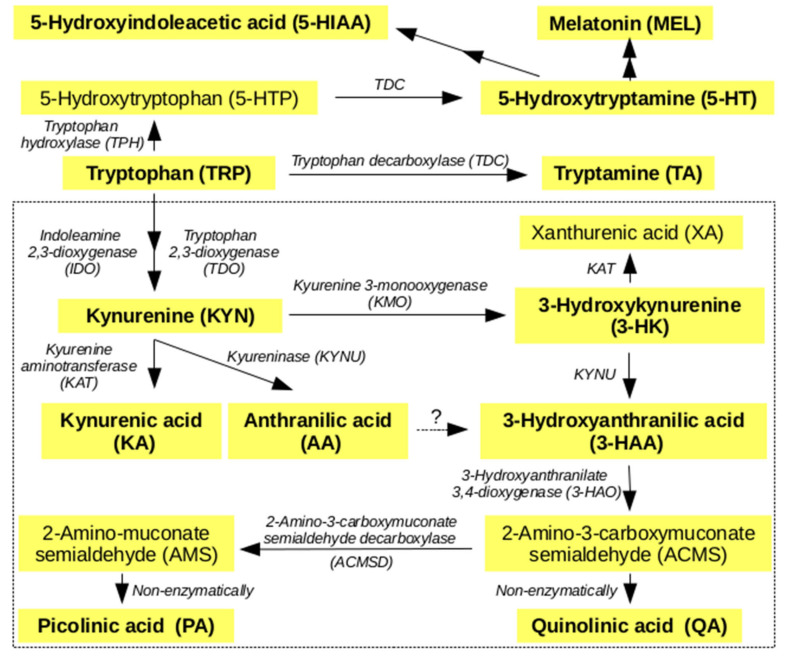
Simpified major metabolic pathways of tryptophan (TRP) in humans. Metabolites marked in bold were quantified by liquid chromatography–tandem mass spectrometry (LC-MS/MS) in the current study. The area enclosed by the dotted line indicates kynurenine pathway (KP).

**Figure 2 metabolites-10-00208-f002:**
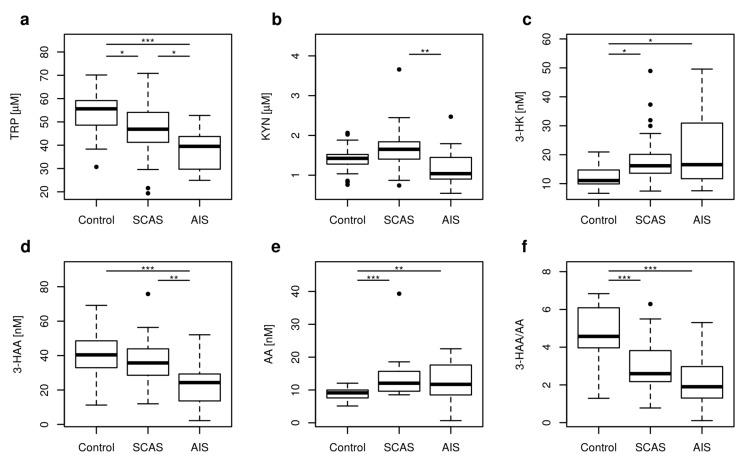
Plasma concentrations of kynurenine pathway metabolites changed in neurovascular disease. The concentrations or ratios of metabolites in the plasma of the controls (control), patients with significant carotid artery stenosis (SCAS), or acute ischemic stroke (AIS) are represented as boxplots: (**a**) TRP: tryptophan; (**b**) KYN: kynurenine; (**c**) 3-HK: 3-hydroxykynurenine; (**d**) 3-HAA: 3-hydroxyanthranilic acid; (**e**) AA: anthranilic acid; (**f**) 3-HAA to AA ratio. Significance codes for post-hoc Dunn test: *** *p* < 0.001, ** *p* < 0.01, * *p* < 0.05. Vertical boxplot is constructed between first (Q1) and third (Q3) quartile, with horizontal median inside. The difference between Q3 and Q1 is called the interquartile range (IQR). Whiskers are drawn from Q1 and Q3 to minimal, respective maximal data point within range (Q1 – 1.5 IRQ) – (Q3 + 1.5 IRQ). Other data are outliers (black circles).

**Figure 3 metabolites-10-00208-f003:**
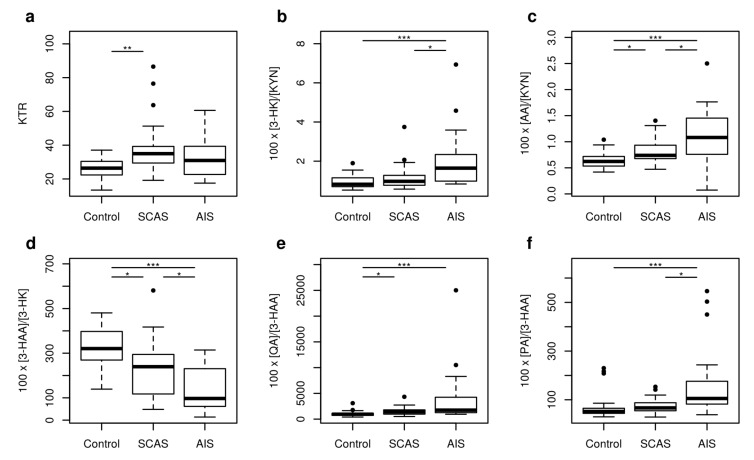
The boxplots of enzyme activities changed in neurovascular disease. (**a**) KTR: IDO activity defined as 1000 × [KYN]/[TRP]; (**b**) 100 × [3-HK]/[KYN]: KMO activity; (**c**) 100 × [AA]/[KYN]: KYNU A activity; (**d**) 100 × [3-HAA]/[3-HK]: KYNU B activity; (**e**) 100 × [QA]/[3-HAA]: 3-HAO activity; (**f**) 100 × [PA]/[3-HAA]: the composed 3-HAO and ACMSD activity. Significance codes for post-hoc Dunn test: *** *p* < 0.001, ** *p* < 0.01, * *p* < 0.05.

**Figure 4 metabolites-10-00208-f004:**
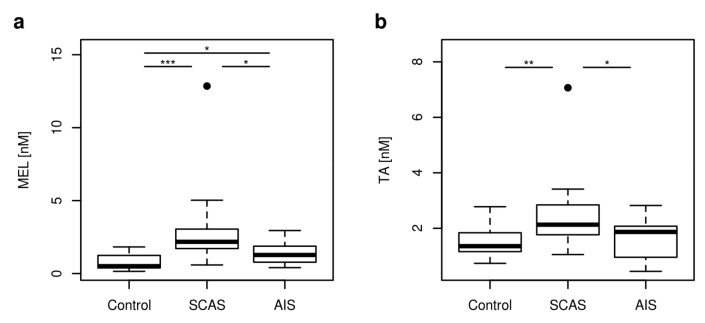
The boxplots of non KP metabolites concentrations changed in neurovascular disease. (**a**) MEL: melatonin; (**b**) TA: tryptamine. Significance codes for post-hoc Dunn test: *** *p* < 0.001, ** *p* < 0.01, * *p* < 0.05.

**Figure 5 metabolites-10-00208-f005:**
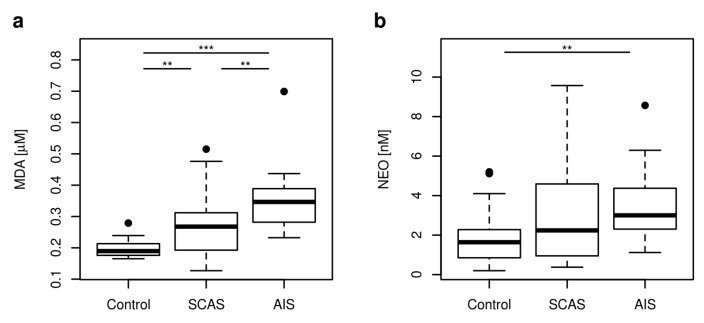
Plasma concentrations of MDA and NEO represented as boxplots. (**a**) MDA concentration was employed as a criterion of OS; (**b**) NEO concentration was employed as a marker of inflammation. Abbreviations: MDA: malondialdehyde; NEO: neopterin. Significance codes for post-hoc Dunn test: *** *p* < 0.001, ** *p* < 0.01.

**Table 1 metabolites-10-00208-t001:** Plasma concentrations of TRP metabolites, NEO, MDA, RBF, and some significant metabolite ratios.

Metabolites	Control (*n* = 25)		SCAS (*n* = 25)		AIS (*n* = 18)		
and Metabolites Ratios	Median	5–95 Percentile	Median	5–95 Percentile	Median	5–95 Percentile	*p*-Value
TRP [μM]	55.62	38.91–68.30	46.87	23.19–61.30	39.51	25.46–51.47	<0.001
KYN [μM]	1.42	0.83–2.00	1.65	0.91–2.43	1.04	0.65–1.89	0.003
3-HK [nM]	11.08	8.22–19.28	16.18	8.09–36.23	16.57	9.71–46.66	0.008
3-HAA [nM]	40.42	17.95–63.90	35.72	15.17–55.66	24.31	3.32–39.17	<0.001
KA [nM]	43.44	25.48–65.99	52.06	21.72–82.21	42.48	21.14–77.63	0.379
AA [nM]	9.13	5.96–11.95	12.06	8.80–18.52	11.71	6.80–21.01	<0.001
QA [nM]	374.60	218.07–492.88	512.61	232.49–753.16	438.44	252.62–648.12	0.065
PA [nM]	24.36	11.84–45.24	23.78	14.47–34.56	23.69	9.05–43.45	0.984
5-HT [nM]	23.71	5.85–60.18	43.49	5.78–134.81	19.92	6.70–86.72	0.263
5-HIAA [nM]	44.05	25.60–111.94	54.64	36.08–91.78	49.48	31.63–77.69	0.102
MEL [nM]	0.51	0.26–1.63	2.17	0.76–4.81	1.27	0.52–2.60	<0.001
TA [nM]	1.36	0.77–2.49	2.13	1.21–3.41	1.87	0.70–2.64	0.001
MDA [μM]	0.19	0.17–0.24	0.27	0.16–0.46	0.35	0.24–0.48	<0.001
NEO [nM]	1.64	0.33–4.91	2.24	0.54–6.93	3.00	1.17–6.63	0.004
RBF [nM]	7.34	4.01–39.21	10.52	3.99–41.02	12.62	5.83–37.92	0.126
3-HAA/AA	4.57	2.26–6.81	2.60	1.02–5.33	1.90	0.45–4.48	<0.001
KTR	26.41	15.18–36.66	34.97	24.56–73.90	30.94	19.02–50.48	0.003
100 × 3-HK/KYN	0.82	0.58–1.51	0.97	0.66–2.04	1.64	0.84–4.93	<0.001
100 × KA/KYN	3.11	2.16–4.66	3.17	1.49–5.34	3.44	2.15–8.78	0.275
100 × AA/KYN	0.62	0.44–0.93	0.74	0.57–1.26	1.08	0.57–1.87	<0.001
100 × 3-HAA/3-HK	320.65	182.70–465.81	239.45	85.77–403.39	97.04	21.29–293.86	<0.001
100 × QA/3-HAA	953.89	448.87–1738.37	1410.83	578.12–2677.36	1745.90	1012.07–12,676.51	<0.001
100 × PA/3-HAA	51.97	35.32–215.07	66.62	37.65–137.12	105.34	49.73–509.71	<0.001

The values are presented as medians with 5th–95th percentiles. A nonparametric Kruskal–Wallis test was used to examine the differences across all subgroups of patients at *p* < 0.05. When the Kruskal–Wallis test was statistically significant, a post hoc Dunn test was performed. Differences between groups based on Dunn test are showed in the following boxplots. Abbreviations: TRP: tryptophan; KYN: kynurenine; 3-HK: 3-hydroxykynurenine; 3-HAA: 3-hydroxyanthranilic acid; KA: kynurenic acid; AA: anthranilic acid; QA: quinolinic acid; PA: picolinic acid; 5-HT: 5-hydroxytryptamine; 5-HIAA: 5-hydroxyindoleacetic acid; MEL: melatonin; TA: tryptamine; MDA: malondialdehyde; NEO: neopterin; RBF: riboflavin; KTR: 1000 × [KYN]/[TRP].

**Table 2 metabolites-10-00208-t002:** Patients’ characteristics.

Patients’ Characteristics	SCAS	AIS	*p*-Value
Number of patients	25	18	-
Sex female/male	8/17	11/7	0.071 ^2^
Arterial hypertension	21	15	>0.999 ^2^
Diabetes mellitus	12	3	0.052 ^2^
Active smoking	12	3	0.052 ^2^
Age ^1^	70 (59.0–78.6)	71 (46.8–82.6)	>0.999 ^3^
BMI (mean ± SD)	28 ± 6	29 ± 8	0.653 ^3^
GF (mean ± SD)	1.31 ± 0.25	1.25 ± 0.36	0.522 ^3^

SCAS: significant carotid artery stenosis; AIS: acute ischemic stroke; BMI: body mass index; GF: glomerular filtration (mL/s); SD: standard deviation. ^1^ Median age; 5th–95th percentiles in parentheses. ^2^ Fisher exact test, ^3^ unpaired t-test.

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
