# Peer review of "Tryptophan Metabolism, Inflammation, and Oxidative Stress in Patients with Neurovascular Disease"

_metabolites, 2020, doi:10.3390/metabo10050208_

Round 1

Reviewer 1 Report

The Authors describe the role of tryptophan metabolism and its role in inflammation and oxidative stress in patients with acute ischemic stroke or with a significant carotid artery stenosis and healthy controls.

They conclude that trypthophan metabolism is deregulated in patients with acute stroke compared to patients with significant carotid stenosis and that inflammation and oxidative stress is enhanced both in acute ischemic stroke and carotid artery stenosis.

However, the article in my opinion is too long especially in the introduction and in the discussion e therefore difficult to read.

Moreover, the final clinical message is unclear.

I suggest that the authors shorten the introduction and the discussion and give greater emphasis on the few positive results suggesting an increased inflammation and oxidative stress in in acute ischemic stroke and carotid artery stenosis patients.

Author Response

Dear Editor, Reviewers, 

Thank you very much for your questions and comments.

I suggest that the authors shorten the introduction and the discussion and give greater emphasis on the few positive results suggesting an increased inflammation and oxidative stress in in acute ischemic stroke and carotid artery stenosis patients.

======================================

1. Yes, we should try to shorten the Introduction and Discussion with an accent on differences between patients with acute ischemic stroke and ischemic carotides in a background of tryptophan metabolism, inflammation and oxidative stress. Perhaps the work would be a bit clearer after omitting metabolites of the serotonin pathway.

2. We tried to explain the links  between the biological processess in a study and the corresponding biomarkers in the aim of study in a condensed form summarizing information about TRP metabolites in inflammation and OS given in introduction.

Lines 102-109: "

Since TRP catabolism was recognized as an important player in inflammation and its metabolites with either prooxidant and antioxidant properties should have relationship with OS, we aimed in this study compare similarities and differences in TRP metabolism, inflammation, and OS in patients with acute ischemic stroke (AIS), more rarely studied significant carotid artery stenosis (SCAS) and in healthy controls. The blood plasma concentrations of both KP and serotonin pathway metabolites (TRP, KYN, 3-HK, 3-HAA, KA, AA, QA, PA, 5-HT, 5-HIAA, MEL), tryptamine (TA), markers of inflammation (NEO, KTR), OS marker MDA, and RBF as one cofactor of enzymes in KP were analyzed and their mutual correlations in studied groups of probands were explored."

3. We attempted rewrite the last part of discussion in more condensed form with emphasis on most important results of study. Lines 324-333.

"In conclusion, we revealed significant differences between SCAS and AIS patients in TRP metabolism, inflammation and OS. For the first time, it was found that the 3-HAA/AA ratio in SCAS patients was reduced compared to controls and was comparable with values for AIS patients. OS in AIS patients was significantly higher compared to SCAS patients and controls. Nevertheless, no correlation was found between the marker of OS MDA and metabolites of TRP in the study groups, except for a negative correlation with antioxidant KA and the activity of KAT in AIS patients. Moderate correlations of TRP metabolites with NEO and KTR in patients with SCAS and AIS clearly supported the relationship of inflammation and TRP metabolism. The results showed that TRP metabolism was more deregulated in AIS compared to SCAS patients and that the effect of TRP metabolites on OS should be further clarified."

4. We added correlation graphs of tryptophan metabolites with markers of inflammation and oxidative stress to more clearly present our results.

Reviewer 2 Report

Comments on Metabolites-789567

In their manuscript ‘Tryptophan Metabolism, Inflammation and Oxidative Stress in Patients with Neurovascular Disease’ Martin Hajsl and colleagues have performed a metabolomic analysis using plasma samples that were collected from healthy controls (n=25) and patients with carotid artery stenosis (n=25) or acute ischemic stroke (n=18). Focus was on tryptophan metabolites, riboflavin, neopterin (marker of inflammation) and malondialdehyde (marker of OS). Based on the results, 7 out of 12 tested tryptophan metabolites were significantly different between studied groups. Malondialdehyde concentrations were higher in acute ischemic stroke and carotid artery stenosis patients compared to controls, and neopterin in acute ischemic stroke group. Malondialdehyde negatively correlated with kynurenine acid and kynurenine aminotransferase (measured as the ratio of kynurenine and kynurenic acid) in acute ischemic stroke group. The manuscript is well written, and the data is presented in a logical way. Clarifying some details, as discussed below, would further improve the manuscript.

Major

  1. Many foods that are rich in tryptophan (poultry, seeds, milk) can also be considered to be healthy in terms of lowering cardiovascular risk. The authors should discuss at least in the limitations section if there is a possibility for underlying connection between dietary tryptophan consumption and risk for carotid artery stenosis and stroke. This might also explain the lower levels of measured tryptophan in patients with carotid artery stenosis and stroke.

  1. The authors clearly demonstrate that there are metabolite differences between plasma samples derived from different patient populations compared to healthy controls. How well do the plasma differences of these metabolites reflect the situation in tissues especially in the ischemic stroke patients? Is it possible that the ischemic core and possibly penumbra are leaking some of the tested metabolites into the bloodstream? Please discuss.

  1. Were carotid artery stenosis patients advised to fast before blood sample collection? If so, please add this detail and discuss.

  1. Table 2 describes patient characteristics followed by p-values. Which statistical test was used to test the differences between the groups? The p-values also have variable amounts of significant digits, for example 0.07, 0.99 and 0.653. Please report them in a consistent manner.

  1. Was the data tested for significant outliers?

Minor

  1. The authors should be more careful with the wording in the sentence: “QA concentrations tended to be higher in patients with SCAS compared to controls (p = 0.064, Dunn test).”, because the p-value was higher than 0.05.

  1. Sentence “Both 5-HIAA and 5-HT plasma concentrations were not significantly different in patients groups of study, although the 5-HT concentration in plasma of patients with SCAS was about twice enhanced as compared with AIS patients or controls” is heavy and needs to be rephrased.

Author Response

Dear Editor, Reviewers, 

Thank you very much for your questions and comments.

  1. Many foods that are rich in tryptophan (poultry, seeds, milk) can also be considered to be healthy in terms of lowering cardiovascular risk. The authors should discuss at least in the limitations section if there is a possibility for underlying connection between dietary tryptophan consumption and risk for carotid artery stenosis and stroke. This might also explain the lower levels of measured tryptophan in patients with carotid artery stenosis and stroke.

The chronic inflammation seems to play the key role in the process of atherogenesis. The effect of diet rich in tryptophan was studied in prospective manner mainly in association with “Mediterranean diet” and tryptophan intake. There was found , than tryptophan level is inversely correlated with cardiovascular disease incidence. Interestingly, there was found no correlation in baseline tryptophan level and it´s 1 year increases with the stroke incidence in opposite to other “non-stroke” cardiovascular events, namely myocardial infarction. When studied only at baseline, the inverse correlation between the tryptophan level and stroke outcome is preserved, what we also found in our study. The possible explanation is, that tryptophan level seems to be more the marker of low-grade inflammation than modifiable factor, minimally in the acute stroke patients.

Yu E, Ruiz-Canela M, Guasch-Ferré M, et al. Increases in Plasma Tryptophan Are Inversely Associated with Incident Cardiovascular Disease in the Prevención con Dieta Meditteránea (PREDIMED) Study. J Nutr 2017;147:314-322.

Mangge H, Reininghaus E, Fuchs D. Role of kynurenine pathway in cardiovascular diseases. In: Mittal S, editor. Targeting the broadly pathogenic kynurenine pathway. Cham (Switzerland): Springer International Publishing; 2015. p. 133–43.

We added a mention about dietary tryptophan consumption and risk for carotid artery stenosis and stroke in the limitations section:

Lines 313-314: “There is a possibility of connection between dietary tryptophan consumption and risk for carotid artery stenosis and stroke. However, we had not information about dietary habits of patients.”

  1. The authors clearly demonstrate that there are metabolite differences between plasma samples derived from different patient populations compared to healthy controls. How well do the plasma differences of these metabolites reflect the situation in tissues especially in the ischemic stroke patients? Is it possible that the ischemic core and possibly penumbra are leaking some of the tested metabolites into the bloodstream? Please discuss.

The blood-brain barrier selective permeability plays important role in the comparison of different metabolites plasma levels. It is a part of the study limitations, but when we compare the results of kynurenine in patients with carotid artery stenoses, where this barrier plays no role,  there are still differences in the same direction as in stroke patients. The level of kynurenine with easy blood-brain barrier transmission is decreased in acute stroke patients. Briefly, in the brain tissue the tryptophan degraded to kynurenine can be metabolized to kynurenin acid in astrocytes. The kynurenic acid plays a protective role in central brain tissue. The kynurenic acid transmission through the blood-brain barrier is poor in the opposite to the kynurenine itself. It may be one of possible explanations, why in many studies the strokes are behaving in different way than the other cardiovascular events – see comment No 1.

Badawy AA. Kynurenine pathway of tryptophan metabolism: regulatory and functional aspects. Int J Tryptophan Res 2017;10:1178646917691938.doi: 10.1177/1178646917691938

Varga N, Csapo E, Majlath Z, Ilisz I, Krizbai IA, Wilhelm I, et al. Targeting of the kynurenic acid across the blood-brain barrier by core-shell nanoparticles. Eur J Pharm Sci 2016;86:67. doi: 10.1016/j.ejps.2016.02.012.

Colpo GD, Venna VR, McCullogh LD, Teixeira AL. Systematic Review on the Involment of the Kynurenine Pathway in Stroke: Pre-Clinical and Clinical Evidence. Front Neurol 2019;10:778. doi: 10.3389/fneur.2019.00778.

  1. Were carotid artery stenosis patients advised to fast before blood sample collection? If so, please add this detail and discuss.

The patients were not fasting, same as the stroke patients. The blood collection status reflects the situation in the plasma in “real-life” conditions. If one accept, that there are biological connections between mentioned markers and clinical outcomes (in our study only hypothesis generating), the influence of food intake is a part of these biological processes.

  1. Table 2 describes patient characteristics followed by p-values. Which statistical test was used to test the differences between the groups? The p-values also have variable amounts of significant digits, for example 0.07, 0.99 and 0.653. Please report them in a consistent manner.

Statistical tests were added and the same amounts of significant digits were used.

  1. Was the data tested for significant outliers?

Yes, we tested the data for significant outlier using non-parametric Dixon test. However, we did  not excluded any data.

Minor                                         

  1. The authors should be more careful with the wording in the sentence: “QA concentrations tended to be higher in patients with SCAS compared to controls (p = 0.064, Dunn test).”, because the p-value was higher than 0.05.

We deleted the sentence.

  1. Sentence “Both 5-HIAA and 5-HT plasma concentrations were not significantly different in patients groups of study, although the 5-HT concentration in plasma of patients with SCAS was about twice enhanced as compared with AIS patients or controls” is heavy and needs to be rephrased.

We shortened the sentence: “Both 5-HIAA and 5-HT plasma concentrations were not significantly different in patients groups of study”

Reviewer 3 Report

In this article, Martin Hajsl and colleagues performed a targeted metabolomics approach in plasma of a small sample of patients with neurovascular disease and healthy controls and they evaluated the kynurenine pathway and some additional markers of oxidative stress and inflammation. The general idea is interesting, the sample groups (although small) seem well-defined, the methodology looks appropriate. However, two relevant issues should be addressed:

  • And most important, the whole metabolomic analysis is based on the already published article of Zhu et al. published in 2011 in Analytical Bioanalytical Chemistry (reference 63 of the manuscript). In that article, the authors use 13C and deuterium-labeled internal standards to achieve accurate quantification, but in the current manuscript, a single, non-labeled internal standard (6-F-TRP) is used. The lack of appropriate internal standards and the lack of a complete validation of the method (considering the relevant differences with the original publication) may result in improper quantification. Did the authors validate the method? Where all the parameters of the method other than linearity (accuracy, precision, matrix effects, recovery…) evaluated? The validity of all the data of this manuscript depends on this critical issue. If this has not been done, I consider that the manuscript is not ready for publication.
  • Throughout the paper, the authors mix biological processes (inflammation, oxidative stress) and specific biomarkers (TRP metabolism, riboflavin). In order to make more sense, a specific link between the biological process of study and the corresponding biomarker should be done.  

In addition to that, some comments, recommendations, and things that should be improved are detailed as follows.

Intro

Figure 1 has an excessive number of non-conventional abbreviations and can be difficult to follow for a non-expert on tryptophan metabolism. I strongly recommend using the complete name of the compounds and the abbreviations in parenthesis. The use of abbreviations could be minimized throughout the paper in order to help the flow and understanding.

Line 89: The authors mention cinnabarinic acid as an example of a kyrunerine but they had not mentioned it before when explaining in detail the metabolism of kynurenine.

Lines 95-96, the term “redox active” sound a little bit odd. Please rephrase the sentence with the same idea.

Lines 96-98: More details should be given (ex. the study was performed in 15 humans and the dose was 6 grams)

Lines 99-101: The authors mention that “reliable markers of oxidative stress are carbonylated proteins, malondialdehyde (MDA), 4-hydroxy-2-nonenal and F2-isoprostanes analyzed mostly by LC-MS/MS”, but they measure only MDA to evaluate oxidative stress. Please provide an explanation and justification.

The abbreviation KTR to measure the kynurenine:tryptophan ration seems confusing. I recommend using KYN:TRP ratio instead.

Lines 110-115. The aim of the study should be rewritten more carefully. The correspondence between a biological process (inflammation, oxidative stress) and the corresponding biomarkers (neopterin, malondialdehyde) should be clearly stated.

Results

Table 1 has some overlapping results with the rest of the graphs. There are some issues:

  • It shows overall differences but does not include differences between groups. Please, include them.
  • It expresses the results as mean ± SD but the authors state that the distributions are not normal. If that is the case, the mean is not the most appropriate way to express the values and alternatives should be employed (e.g. median). If this is not done, the interpretation of the data can be confusing. As an example, the 100 x [PA]/[3-HAA] ratio has a mean of 175 in the AIS group which is clearly influenced by three outliers, and the median will probably be 110 (much far form 175) and much closer to the other groups (control and SCAS, with values for this variable of around 70).

The authors perform several correlations (e.g. line 169) but they do not show any of them. The most relevant correlations graphs should be included at least in the supplementary material.

Discussion

The authors mix again biological processes (inflammation, oxidative stress) and specific biomarkers (TRP metabolism, riboflavin). In order to make more sense, a specific link between the biological process of study and the corresponding biomarker should be done.

The specificity of some results in the discussion makes difficult to have a general overview of the purpose and the major conclusions of the paper. A major modification of it trying to simplify the message is encouraged.

The experience of our lab in the analysis of tryptophan metabolites by LC-MS/MS is that plasma serotonin is not a very valuable marker, as most of it (90%) is generally transported in the platelets. Please comment on this and explain whether you consider that circulating 5HT is a good way to measure this compound.

Methods

The authors should explain how they calculated sample size and why the N significantly differs between groups (25 patients with SCAS vs 18 patients with AIS vs 25 healthy controls).

Line 352: mean age

Line 396: MDA eluted at 3.05 min and MetMDA eluted at 3.54 min. The authors used an isocratic method, but they do not specify which is the total duration of the method. Additionally, they performed derivatization of these compounds before analyzing them so the analytes that eluted at 3.05 and 3.54 were not MDA and MetMDA but they derivatized analogs (with DNPH).

Table 2 and the corresponding description is a result rather than a method.

Statistical analysis should explain that boxplots were employed and how they are calculated (some authors use Q1,Q3 and max, whereas other use 5 and 95 percentiles or other approaches).

Author Response

Dear Editor, Reviewers, 

Thank you very much for your questions and comments.

  • And most important, the whole metabolomic analysis is based on the already published article of Zhu et al. published in 2011 in Analytical Bioanalytical Chemistry (reference 63 of the manuscript). In that article, the authors use 13C and deuterium-labeled internal standards to achieve accurate quantification, but in the current manuscript, a single, non-labeled internal standard (6-F-TRP) is used. The lack of appropriate internal standards and the lack of a complete validation of the method (considering the relevant differences with the original publication) may result in improper quantification. Did the authors validate the method? Where all the parameters of the method other than linearity (accuracy, precision, matrix effects, recovery…) evaluated? The validity of all the data of this manuscript depends on this critical issue. If this has not been done, I consider that the manuscript is not ready for publication.

            Validation experiments were added into end of the section 4.4. Tryptophan Metabolites, NEO and RBF Analysis in methods. The results were summarized in Table A2 and A3.

"Validation experiments of the method were performed. The linear range for each analyte was evaluated using a series of diluted calibration mixtures with internal standard added to each sample measured five times. The calibration curves were constructed using ratios of the analytes to internal standard versus the corresponding concentration of analyte. The solutions were prepared in 4% bovine albumine in phoshate buffered saline (Sigma, Czech Republic) treated with 15 mg/ml of active charcoal (Sigma, Czech Republic) overnight at room temperature. Signal to noise (S/N) value for each analyte was evaluated using the script within Analyst software. The LOD for each analyte was defined as concentration of analyte with S/N >= 3. The results are summarized in Table A2.

 Matrix effects were calculated as a percentage of ratio of calibration curve slope in plasma and calibration curve slope in aqeous solution [64]. Six different plasma samples were spiked with analytes in a range of concentrations indicated in Table A2.

The recovery and imprecision was determined in six different plasma samples spiked with calibration mixture at concentrations indicated in Table A3. The recovery was calculated as a percentage of difference of measured and added concentration divided by added concentration."

  • Throughout the paper, the authors mix biological processes (inflammation, oxidative stress) and specific biomarkers (TRP metabolism, riboflavin). In order to make more sense, a specific link between the biological process of study and the corresponding biomarker should be done.  

We tried to explain the links between the biological processess in a study and the corresponding biomarkers in the aim of study in a condensed form summarizing information about TRP metabolites in inflammation and OS given in introduction. Lines 102-109:

"Since TRP catabolism was recognized as an important player in inflammation and its metabolites with either prooxidant and antioxidant properties should have relationship with OS, we aimed in this study compare similarities and differences in TRP metabolism, inflammation, and OS in patients with acute ischemic stroke (AIS), more rarely studied significant carotid artery stenosis (SCAS) and in healthy controls. The blood plasma concentrations of both KP and serotonin pathway metabolites (TRP, KYN, 3-HK, 3-HAA, KA, AA, QA, PA, 5-HT, 5-HIAA, MEL), tryptamine (TA), markers of inflammation (NEO, KTR), OS marker MDA, and RBF as one cofactor of enzymes in KP were analyzed and their mutual correlations in studied groups of probands were explored."

Moreover, recent studies linked the kynurenine pathway with atherosclerosis, namely with immune vessel stability. The inflammation response in the vessel tree is the reaction to the incorporation of oxidized low-density lipoproteins triggering the recruitment of monocytes and T-cells. IDO, one of the key enzymes of tryptophan metabolism, plays a substantial role in atherosclerotic plaque, vessel and immune cells. In the animal models, the genetic ablation of IDO leads to increased vascular inflammation. Increased IDO expression on the other hand protects against atherosclerosis. Persistent low-grade inflammation can lead to the disruption of the cap of atherosclerotic plaque resulting in it´s rupture and promoting thrombosis, which can lead to the fatal complications, namely stroke and myocardial infarction.

Intro

1. Figure 1 has an excessive number of non-conventional abbreviations and can be difficult to follow for a non-expert on tryptophan metabolism. I strongly recommend using the complete name of the compounds and the abbreviations in parenthesis. The use of abbreviations could be minimized throughout the paper in order to help the flow and understanding.

The Figure 1 was redrawn according reviewer’s recommendation

2. Line 89: The authors mention cinnabarinic acid as an example of a kyrunerine but they had not mentioned it before when explaining in detail the metabolism of kynurenine.

The sentence was completed by:
Line 80:  “cinnabarinic acid produced from 3-HAA by catalase as a minor product”

3. Lines 95-96, the term “redox active” sound a little bit odd. Please rephrase the sentence with the same idea.

We substituted the term “redox-active”by:
Line 86-87: ”In addition, high concentrations of kynurenine metabolites may be either prooxidative or antioxidant...”

4. Lines 96-98: More details should be given (ex. the study was performed in 15 humans and the dose was 6 grams)

The sentence was rewritten:
Line 87-90 “As an example, in a study where 15 human subjects were overloaded by oral administration of 6 g TRP induction of OS was observed probably due to the generation of KP metabolites QA, 3-HA, and 3-HAA, all of which known to have the ability to generate free radicals [34].”

5. Lines 99-101: The authors mention that “reliable markers of oxidative stress are carbonylated proteins, malondialdehyde (MDA), 4-hydroxy-2-nonenal and F2-isoprostanes analyzed mostly by LC-MS/MS”, but they measure only MDA to evaluate oxidative stress. Please provide an explanation and justification.

We discussed it in limitations of our study (line 315-316). It seems to us  that MDA measured by LC-MS/MS is a reliable and commonly used marker of OS status.

6. The abbreviation KTR to measure the kynurenine:tryptophan ration seems confusing. I recommend using KYN:TRP ratio instead.

The KTR abbreviation is common in publications dealing with TRP. We would like to maintain KTR as an abbreviation for 1000*[KYN]/[TRP].

7. Lines 110-115. The aim of the study should be rewritten more carefully. The correspondence between a biological process (inflammation, oxidative stress) and the corresponding biomarkers (neopterin, malondialdehyde) should be clearly stated.

We tried to explain the links between the biological processess in a study and the corresponding biomarkers in the aim of study in a condensed form summarizing information about TRP metabolites in inflammation and OS given in introduction. Lines 102-109.

Results

8.  Table 1 has some overlapping results with the rest of the graphs. There are some issues:

  • It shows overall differences but does not include differences between groups. Please, include them.

Due to insufficient space for extended table,  the differences between groups are shown in boxplots

  • It expresses the results as mean ± SD but the authors state that the distributions are not normal. If that is the case, the mean is not the most appropriate way to express the values and alternatives should be employed (e.g. median). If this is not done, the interpretation of the data can be confusing. As an example, the 100 x [PA]/[3-HAA] ratio has a mean of 175 in the AIS group which is clearly influenced by three outliers, and the median will probably be 110 (much far form 175) and much closer to the other groups (control and SCAS, with values for this variable of around 70).

The Table 1 was rearranged according to reviewer’s proposal. The measured values are presented as medians with 5th–95th percentiles. 

9. The authors perform several correlations (e.g. line 169) but they do not show any of them. The most relevant correlations graphs should be included at least in the supplementary material.

Three graphs with correlations were added to the manuscript (Figure B1, B2, B3)

Discussion

10. The authors mix again biological processes (inflammation, oxidative stress) and specific biomarkers (TRP metabolism, riboflavin). In order to make more sense, a specific link between the biological process of study and the corresponding biomarker should be done.

It is a bit complicated. As an example, it was shown 3-HK, 3-HAA and QA can induce apoptosis of T cells, monocytes and epithelial cells. 3-HAA is referred to as an anti-inflammatory metabolite influencing the release of several interleukins. [1,2] Kynurenine modulates the endothelial response, e.g. adhesion of monocytes under flow conditions and nitric oxide production [3]. In experiment the KP modulates the inflammatory response in vascular and immune cells. In the plaque IDO expression was confirmed in several types of plaque cells [4]. Extrahepatic degradation of TRP becomes significant only when IDO expression is increased, namely in response to inflammation. Downstream of IDO in the KP, the levels of 3-HK and 3-HAA have been associated with inflammation and cardiovascular disease in patients with severe renal dysfunction [5].

  1. Mailankot M, Nagaraj RH. Induction of indoleamine 2,3-dioxygenase by interferon-gamma in human lens epithelial cells: apoptosis through the formation of 3-hydroxykynurenine. Int J Biochem Cell Biol 2010;42:1445-1454.
  2. Barth MC, Ahluwalia N, Anderson TJ et al. Kynurenic acid triggers firm arrest of leucocytes to vascular endothelium under flow conditions. J Biol Chem 2009;284:19189-19195.
  3. Wang Y, Liu H, McKenzie G et al. Kynurenine is an endothelium-derived relaxing factor produced during inflammation. Nat Med 2010;16:279-285.
  4. Niinisalo P, Oksala N, Levula M et al. Activation of indoleamine 2,3-dioxygenase-induced tryptophan degradation in advanced atherosclerotic plaques: Tampere Vascular Study. Ann Med 2009;42:55-63.
  5. Pawlak K, Mysliwiec M, Pawlak D. Kynurenine pathway – a new link between endothelial dysfunction and carotid atherosclerosis in chronic kidney disease patients. Adv Med Sci 2010;55:196-203.

11. The specificity of some results in the discussion makes difficult to have a general overview of the purpose and the major conclusions of the paper. A major modification of it trying to simplify the message is encouraged

We attempted rewrite the last part of discussion in more condensed form with emphasis on most important results of study.

"In conclusion, we revealed significant differences between SCAS and AIS patients in TRP metabolism, inflammation and OS. For the first time, it was found that the 3-HAA/AA ratio in SCAS patients was reduced compared to controls and was comparable with values for AIS patients. OS in AIS patients was significantly higher compared to SCAS patients and controls. Nevertheless, no correlation was found between the marker of OS MDA and metabolites of TRP in the study groups, except for a negative correlation with antioxidant KA and the activity of KAT in AIS patients. Moderate correlations of TRP metabolites with NEO and KTR in patients with SCAS and AIS clearly supported the relationship of inflammation and TRP metabolism. The results showed that TRP metabolism was more deregulated in AIS compared to SCAS patients and that the effect of TRP metabolites on OS should be further clarified."

12. The experience of our lab in the analysis of tryptophan metabolites by LC-MS/MS is that plasma serotonin is not a very valuable marker, as most of it (90%) is generally transported in the platelets. Please comment on this and explain whether you consider that circulating 5HT is a good way to measure this compound.

13. We tried to show whether 5-HT is released from platelet on atherosclerotic plaques in SCAS patients. We found non significant emhancement of 5-HT concentrations in SCAS patients (Table 1).

Methods

14. The authors should explain how they calculated sample size and why the N significantly differs between groups (25 patients with SCAS vs 18 patients with AIS vs 25 healthy controls).

This was planned as an observational pilot study based on clinical definition of the groups, reflecting the biological differences between atherosclerosis (SCAS), atherothrombosis (AIS) and healthy controls and it´s potential association with the markers of KP and oxidative stress. The number of patients reflects the real life numbers of patients enabling us to match groups in the respect for the risk factors to minimize the bias – see Table 2.

15. Line 352: mean age

Lines 355-356: The sentence was rewritten: “The median age of the controls was 59 (38.2-65.0) ( 5th–95th percentiles in parentheses) ,...“

16. Line 396: MDA eluted at 3.05 min and MetMDA eluted at 3.54 min. The authors used an isocratic method, but they do not specify which is the total duration of the method. Additionally, they performed derivatization of these compounds before analyzing them so the analytes that eluted at 3.05 and 3.54 were not MDA and MetMDA but they derivatized analogs (with DNPH).

The information of the time of analysis was added with correct names of analytes:

Lines 398-399:  “DNPH derivatives of MDA and MetMDA eluted at 3.05 and at 3.54 min, respectively; total time of analysis was 6.5 min”.

17. Table 2 and the corresponding description is a result rather than a method.

We prefer to keep Table 2 in Methods, near to information about blood drawing.

18. Statistical analysis should explain that boxplots were employed and how they are calculated (some authors use Q1,Q3 and max, whereas other use 5 and 95 percentiles or other approaches).

In the legend for the first boxplot (Figure 2) we added the explanation of boxplots construction.

Lines 143-146: “Vertical boxplot is constructed betveen first (Q1) and third (Q3) quartile, with horizontal median inside. The difference between Q3 and Q1 is called the interquartile range (IQR). Whiskers are drawed from Q1, and Q3 to minimal, respective maximal data point within range (Q1 – 1.5 IRQ) – (Q3 + 1.5 IRQ). Other data are outliers.”

Round 2

Reviewer 1 Report

The authors have done the requested improvements

Reviewer 3 Report

The authors have addressed in a timely and appropriate way all the comments. Additionally, they have extensively modified several parts of the manuscript accordingly, and they have included the validation parameters (which was the major concern of the previous review).

I found the manuscript is very interesting, well-written and the results are very nice. I consider that they have done a very nice piece of work which would be of interests to the readers. 

ps: If you have the chance, it would be good to make sure that some commas that appear in the new tables that have the validation results could be replaced by points (ex. 6,25 should be 6.25 for consistency).